# Exploratory Metabolomic Fingerprinting of Aqueous Humor in Healthy Horses and Donkeys, and in Horses with Ocular Pathologies

**DOI:** 10.3390/ani15192810

**Published:** 2025-09-26

**Authors:** Ignacio Corradini, Eduard Jose-Cunilleras, Pau Nolis, María Mar López-Murcia, Aloma Mayordomo-Febrer

**Affiliations:** 1Department of Animal Medicine and Surgery, Facultat de Veterinària, Universitat Autònoma de Barcelona, 08193 Bellaterra, Barcelona, Spain; ignacio.corradini@gmail.com (I.C.); eduard.jose.cunilleras@uab.cat (E.J.-C.); 2School of Veterinary Medicine and Science, The University of Nottingham, Sutton Bonington LE12 5RD, UK; 3Nuclear Magnetic Resonance Facility, Universitat Autònoma de Barcelona, 08193 Bellaterra, Barcelona, Spain; pau.nolis@uab.cat; 4Departamento de Medicina y Cirugía Animal, Facultad de Veterinaria, Universidad Cardenal Herrera-CEU, CEU Universities, Tirant lo Blanc, 7, 46115 Alfara del Patriarca, Valencia, Spain; mlopez@uchceu.es

**Keywords:** metabolomics, fingerprinting, aqueous humor, horses, donkeys, oculomics

## Abstract

This study aimed to characterize the aqueous humor metabolome in healthy horses and donkeys, and to investigate whether age, sex or ocular lesions in horses might be associated with metabolic tendencies in aqueous humor using nuclear magnetic resonance spectroscopy. Some metabolites showed patterns of variation linked to age, sex and ocular disease. These findings should be interpreted with caution due to the relatively small and heterogenous samples sizes; however, they highlight potential metabolomic differences that could guide future targeted studies in larger, stratified cohorts. Our work provides preliminary insight into the aqueous humor metabolome and a methodological framework for using nuclear magnetic resonance spectroscopy metabolomics in equid aqueous humor.

## 1. Introduction

Metabolites are small organic molecules involved in various biochemical processes within an organism. The concentrations of these metabolites can provide valuable insights into the physiological and pathological conditions of biological systems [1]. Clinical metabolomics analysis has been used in biomedical research for several objectives, including the evaluation of metabolic pathways in health and disease, as well as the discovery of new potential biomarkers of disease [2,3,4,5,6,7,8].

A significant portion of metabolomics research has focused on blood (serum and plasma) and urine to identify disease biomarkers in humans and other species, including horses [9,10,11,12,13,14,15]. However, increasing attention is being directed towards studying the metabolomic profiles and characteristics of various other biofluids, which may exhibit local or regional differences compared to serum or urine. This expanded approach offers promising opportunities not only for identifying specific disease biomarkers but also for investigating metabolic pathways that may be altered in various disease states. Numerous studies have explored the metabolomic profiles of different biofluids, including articular fluid, peritoneal effusion, cerebrospinal fluid, saliva, seminal plasma, and exhaled breath condensates, among others [16,17,18,19,20,21,22].

In the context of ocular health, analyzing metabolites in aqueous humor (AH) can provide valuable insights into eye diseases. The metabolic profile of human AH has been documented, which is an important step in comparing the metabolomes of eyes with specific diseases. This approach will help identify altered metabolic pathways, possible pharmacological targets, as well as potential biomarkers of disease [23,24,25,26,27,28].

In the developed world, the relationship between horses and humans has evolved. Nowadays, in addition to work and transport, horses are also used for competition and pleasure riding. Vision plays a critical role in all these activities. Therefore, scientific interest in equine ophthalmology is increasing, leading to more research. While the metabolomic profile of AH has been studied in rabbits, humans and rats, it has not yet been evaluated in horses or donkeys [23,24,29,30].

The first objective of this research was to generate foundational metabolomic data of AH in horses and donkeys, using nuclear magnetic resonance (NMR) spectroscopy. A secondary goal was to explore potential metabolomic differences between males and females, as well as across different age groups of horses. Another secondary objective was to explore potential differences in the metabolomic composition of AH of healthy horses and that of horses with naturally occurring ocular disorders.

## 2. Materials and Methods

### 2.1. Animals and Study Design

This study was designed as a prospective, observational, exploratory investigation aimed at characterizing the metabolomic profile of the AH in equids, including both healthy horses and those with naturally occurring ocular lesions. In addition, we had the opportunity to obtain AH from three donkeys. However, due to the limited sample size, a description of the metabolomic signature is provided but excluded from the statistical analysis. The horses (N = 24) and donkeys used in this study were animals destined for slaughter and acquired by the “Masked for review” to carry out terminal anesthesia workshops unrelated to ocular disease. Inclusion criteria required a normal complete blood cell count and comprehensive biochemical profile, as well as an unremarkable full physical examination. All animals enrolled in the study presented a body condition score between 4 and 6 out of 9. Only animals that were not receiving any systemic or topical medication were enrolled. In addition, all animals were appropriately vaccinated and dewormed in accordance with standard protocols, as these were prerequisites for admission and continued housing in the facilities. Horses that did not meet any of these health or management criteria were excluded from the study.

Animals were housed at “Masked for review” for a period of one to two weeks until the terminal workshops took place. During this time, all animals were provided with a uniform diet consisting of ad libitum hay and equal proportions of alfalfa cubes and maintenance commercial concentrate. They were kept under identical conditions, with access to the same water source, and were housed in indoor stalls with continuous access to small individual paddocks.

All procedures involving animals were conducted in accordance with the guidelines and regulations of “Masked for review” and were approved by the Ethics Committee of this institution.

### 2.2. Ophthalmic Examination

The ocular examination was performed 24 h prior to sample collection. The examination included vision assessment (menace response, dazzle and pupillary light reflexes), evaluation of corneal and palpebral reflexes, slit-lamp examination of the anterior segment (Kowa SL-15^®^, Kowa Optimed, Düsseldorf, Germany), intraocular pressure measurement using a rebound tonometer (TonoVet iCare^®^, Icare Finland Oy, Vantaa, Finland) and fluorescein staining to assess the integrity of the corneal epithelium. The ocular fundus was evaluated using a PanOptic ophthalmoscope (PanOptic^®^, WellchAllyn, Norfolk, UK) and a direct ophthalmoscope (WellchAllyn, Norfolk, UK).

### 2.3. Sample Collection and Preparation

Five hundred microliters of AH were obtained within 5 min after humane euthanasia, through para-aquo-centesis at the scleral–corneal junction using a 27G 0.5-inch needle attached to a 1 mL syringe. Samples were immediately placed on ice and transferred to a −80 °C freezer within 20 min of collection to preserve metabolite integrity until analysis. Due to logistical issues related to the anesthesia workshop, it was not possible to obtain samples from both eyes in 9 horses because the head could not be turned within five minutes postmortem. However, AH was successfully obtained from both eyes of all donkeys.

Before NMR spectra acquisition, sample preparation involved mixing 455 μL of AH with 45 μL of the NMR buffer solution, which consisted of Deuterium oxide (D_2_O), sodium-3′-trimethylsilylpropionate-2,2,3,3-d4 (TSP), and Calcium phosphate buffer (with a TSP final concentration within the NMR tube of 0.6 mM). The prepared sample was then placed in a 5 mm high-field NMR glass tube and centrifuged to remove any air bubbles.

### 2.4. Data Acquisition and Analysis

The NMR spectra were acquired under standardized conditions at the “masked for review”. NMR spectra were obtained using a Bruker Avance III DRX 600 spectrometer (Bruker Biospin GmbH, Rheinstetten, Germany) equipped with a triple resonance 1H/13C/31P probe, operating at a 1H frequency of 600.13 MHz. The sample temperature was maintained at 310 K throughout acquisition. To minimize artifacts from water suppression and to selectively detect signals from metabolites with long T_2_ relaxation times, we employed the Long-Time CPMG pulse sequence, which incorporates a T_2_ filter set to 325 ms immediately after water suppression. For each sample, 256 free induction decays (FIDs) were recorded, each with 64k data points spanning a spectral width of 14 ppm and using a recycling delay of 1 s.

### 2.5. Quantification of Metabolites

The acquired spectra were processed using the TopSpin software (v4.4.1, Bruker, Rheinstetten, Germany) for phase and baseline correction. After correction, the spectra were loaded into the software Chenomx NMR suite 8.3, specifically utilizing the Processor module for spectra pre-processing and the profiler module for metabolite identification and quantification. Signal integration with fitting methods was employed to determine the peak areas or volumes corresponding to specific metabolites (https://www.chenomx.com/). Reference compound 0.6 mM TSP was used as an internal standard for the accurate quantification of metabolite concentrations.

### 2.6. Comparative Analysis

Due to the exploratory nature of this study, a comparative analysis was performed by dividing normal horse eyes by sex (male and female) and into two age groups (young and adult horses between 1 and 15 years old and senior and geriatric horses older than 16 years old). This grouping strategy was chosen to enable the detection of preliminary patterns that could inform future, more powerful hypothesis-driven studies.

Horse eyes with ocular lesions were divided into three groups: eyes with anterior chamber disease (ACD), eyes with cataracts, and eyes with retinal lesions. Each disease group was analyzed as a whole category; however, further description of the conditions within each group was provided separately.

Data are summarized by descriptive statistics as median and interquartile range (IQR) (mM). To assess trends in AH metabolite concentration differences between groups, we applied linear mixed-effects models using the lmer function from the lme4 package in R (v4.2.1). For each metabolite, a univariable mixed model was constructed with horse ID as the random effect to account for within-subject correlation, as some animals contributed samples from both eyes. Fixed effects included group comparisons with the control group as reference.

As this is an exploratory study, we applied a two-stage approach for significance testing and false discovery rate (FDR) control. First, we retained metabolites for group comparisons with a raw *p*-value of <0.2. We used this threshold to increase sensitivity while preserving robustness in the context of high biological variability with a moderate sample size. These group comparisons were then subjected to multiple testing corrections using the Benjamini–Hochberg FDR method. We only report metabolite differences as statistically significant if, after adjusting for FDR, the FDR-adjusted *p*-value was <0.05, and we present the differences as trends if the FDR-adjusted *p*-value falls between >0.05 and <0.2, meriting further validation.

To identify potential latent structures within the dataset, principal component analysis (PCA), partial least squares discrimination analysis (PLDSA) and random forest analysis were performed using the MetaboAnalyst online platform (v6.0) developed by the Wishart Research Group [31]. The data was normalized by a pooled sample using the control group as reference, square root transformation was applied, and data was auto-scaled using the mean-centered divided by the standard deviation of each variable. This multivariate analysis was used strictly for exploratory purposes to identify potential latent structures within the dataset and not for inference as this is an exploratory study. These methods were used to ensure that univariate findings were not artefacts of group stratification or high-dimensional noise.

## 3. Results

### 3.1. Samples

In total, 45 AH samples were obtained, 39 from horses and 6 from donkeys. Finally, samples from 40/45 eyes were included in the study. The remaining five AH samples were excluded for two reasons: four horse samples had a poor NMR-spectra baseline shape, and one was from the only donkey with ocular pathology (cataract).

Of 40 AH samples included in the study, 35 were obtained from equine eyes and 5 from asinine eyes. As mentioned before, all asinine samples corresponded to healthy eyes (asinine group N = 5). Regarding horses, 17 samples were from eyes without ocular pathology (equine control group N = 17) of which 8 belonged to horses younger than 15 years old and 9 to horses aged 16 years or older. Regarding sex distribution, eight were control eyes from females and nine were control eyes from males (five eyes from geldings and four from stallions). Due to the limited number of donkey eyes, comparisons with horses, and between donkey’s sex and age groups were not performed.

Regarding AH samples from eyes with ocular pathology; eight corresponded to eyes with cataracts (cataract group N = 8), six to eyes with retinal disease (retina group N = 6) and four to eyes with ACD (ACD group N = 4). Detailed descriptions of the ocular lesions within each group are provided in Table 1.

### 3.2. Characterization of the Metabolomic Profile of the AH of the Horse and Donkey Healthy Eyes

A total of 27 metabolites were identified and quantified in AH from healthy eyes of horses and donkeys. Most metabolites were identified in all samples; however, there were instances in which some metabolites were not conclusively identified and were excluded from the analysis (Table 2). A total of 13 out of the 27 metabolites were amino acids (alanine, arginine, glutamine, histidine, isoleucine, leucine, phenylalanine, proline, serine, threonine, tryptophan, tyrosine and valine), 2 were amino acid derivatives (creatine and creatinine), 3 were alcohols (ethanol, methanol and propyleneglycol), 5 were organic acids (acetate, citrate, 2-hydroxyisovalerate, lactate and pyruvate) and finally 1 was a saccharide (glucose), an amide (urea), an organic compound (dimethyl sulfone) and a vitamin (ascorbate). Most metabolites were present at similar concentrations in the AH of both horses and donkeys, indicating a relatively conserved metabolic milieu across both species (Table 2 and Figure 1). Figure 2 shows the NMR spectra of the AH of one of the horses with relevant metabolites assigned.

### 3.3. Characterization of the Metabolomic Profile of AH According to Age and Sex Groups of Normal Equine Eyes

The mean age of the older adults and senior group was 20.7 (range 16–31) years of age, while the younger horses had a mean age of 10.9 (range 2–15). Threonine levels were significantly higher in young and young adult horses (0.14 ± 0.08 mM) compared to older adults and seniors (0.07 ± 0.08 mM), * *p* = 0.04. The concentrations of the other metabolites were similar across both age groups. Creatine levels were significantly higher in males (0.08 ± 0.03 mM) compared to females (0.04 ± 0.02 mM), * *p* = 0.04.

### 3.4. Characterization of the Metabolomic Profile of the AH of the Equine Eyes with Ocular Disease

PCA score plots (Figure 3) comparing control samples to those with cataract, retina, and ACD revealed no extreme outliers or batch effects. While some separation tendencies were observed, these patterns were not robust and were not used for inferential purposes. As recommended for small exploratory datasets, PCA was employed solely to visualize potential clustering and data distribution, without drawing statistical conclusions.

Compared to the control group (0.12 ± 0.07 mM) arginine was significantly lower in the retina group (0.07 ± 0.01 mM, * *p* = 0.05). On the other hand, dimethyl sulfone was significantly higher in the retina (0.28 ± 0.21 mM, * *p* < 0.00) and cataract (0.16 ± 0.23 mM, * *p* = 0.05) groups compared to the control group (0.07 ± 0.05 mM). In the same way of arginine, valine was significantly lower in the retina group (0.15 ± 0.07 mM, * *p* = 0.03) compared to the control group (0.50 ± 0.33 mM) (Table 3).

Several other metabolites showed trends to differentiate between the control group and groups with eye disease, though these did not reach significance after FDR correction. Compared to the control group, arginine exhibited a trend toward lower concentration in the cataract group, ascorbate was lower in both the ACD and retina groups, creatinine concentration was lower in the retina group, and 2-hydroxivalerate showed a trend toward higher concentration in the cataract group. Arginine (*p* = 0.06) was reduced in the cataract group compared to controls. Ascorbate (*p* = 0.12) was lower in both the ACD and retina groups relative to controls. Pyruvate (0.09 mM) concentration was higher in the ACD group relative to the control group. Lastly, valine (0.09 mM) concentration was lower in the ACD group compared to controls (Table 4).

## 4. Discussion

This foundational exploratory study characterizes the most abundant metabolites present in the AH of horses and donkeys (Table 2 and Figure 1). These results are particularly valuable as they represent the first reported metabolomic signature of the AH in these species, providing a novel reference framework for future studies on equid ocular physiology and pathology. The similarities between the metabolomic signature of the AH of both species are not particularly surprising, as they may extend to other mammals as well [29]. Studies in different species often show a relatively conserved metabolic profile in the AH, with common metabolites such as glucose, lactate, amino acids, and organic acids being present at similar concentrations [23,24,29,30]. The consistency in metabolite concentrations across these species reflects the similar physiological roles of AH in nourishing avascular tissues like the lens and cornea, as well as maintaining intraocular pressure.

The AH is produced by the ciliary body, in which both active secretion and passive ultrafiltration of plasma are involved. Research comparing the metabolomic profiles of plasma and AH in humans has identified a shared core metabolome across these two biofluids, although differences in their concentrations have been recognized [23]. A prior study in horses examining the metabolomic profiles of plasma, urine and feces found at least 14 core metabolites common to all three biofluids [13]. Although plasma metabolomics were not concurrently analyzed in the current study, the 14 core metabolites from Escalona’s study [13] were also detected in the AH of both horses and donkeys.

### 4.1. Age-Related Variations in AH Composition

Our results suggest that AH threonine concentration in horses varies with age, with older horses showing statistically significantly lower concentrations than younger ones. Threonine, an essential amino acid, plays a crucial role in protein synthesis, collagen production, and maintaining tissue integrity, particularly in connective tissues, like the cornea and lens [32]. Age-related metabolic changes, including reduced amino acid absorption, altered protein metabolism and decreased intake in aging subjects, are well-documented in humans [33]. This finding could point to a potential link with metabolic changes associated with aging. Interestingly, while age-related metabolic shifts are widely studied in human medicine, there is a lack of specific data on threonine concentrations in the AH of horses or other species. This warrants further investigation to understand the implications of amino acids in ocular aging fully. Confirming whether these changes are unique to threonine or extend to other amino acids could lead to a broader understanding of aging in equine ocular tissues.

### 4.2. Significant Metabolite Changes in Horse AH of Eyes with Cataracts and Retinal Disease

Significant decreases in arginine and valine concentrations were observed in the AH of horses of the retina group. Moreover, arginine exhibited a decreasing trend in the cataract group, while valine showed a similar trend in the ACD group compared to the control group.

Research involving human subjects has yielded promising results in the identification of potential biomarkers of retinal disease in AH [34,35,36], and similar reductions in arginine have been identified in AH of eyes with age-related nuclear cataracts [37]. Arginine is an essential precursor of nitric oxide and polyamines [38], which have antioxidant effects, and its metabolism is critical for photoreceptors’ health [39]. Impaired availability of arginine in AH may lead to decreased antioxidant activity in the lens and other structures within the eye. It has been hypothesized that ciliary body dysfunction and lens epithelial dysfunction decrease active transport of arginine and other antioxidants into the AH and the lens [37]. This could exacerbate oxidative stress, promoting covalent modification and unfolding of lens protein, which is a well-documented mechanism of cataractogenesis [40]. However, physiological functions, and production and breakdown of arginine in the eye are difficult to analyze due to the compartmentalized expression and interaction with other factors [41].

Valine has also been reported to have decreased in the AH of human patients with cataracts. Branched-chain amino acids, such as valine, have a relevant role in protein synthesis and energy metabolism [42]. In a previous study, valine and other branched chain amino acid biosynthesis pathways were shown to be altered in the AH of patients with retinal vein occlusion macular oedema [43]. Furthermore, an experimental study demonstrated that valine was significantly reduced in the vitreous humor of pigs with induced photoreceptor degeneration compared to controls. The authors hypothesized that decreased photoreceptor activity could lead to a reduction in amino acid levels, as exemplified by the reduction in valine [43,44]. Further studies in horses are warranted to elucidate the role of altered amino acid metabolism and changes in AH metabolite concentrations associated with retinal disease.

Dimethyl sulfone showed a significant increase in the cataract and retina groups compared to the control group; notably, in the retina group, it showed a four-fold increase in absolute terms. Dimethyl sulfone is considered an endogenous “co-metabolite”, arising from the combined metabolism of sulphur-containing compounds by the host and the microbiota [45] with proposed links to mitochondrial regulation in horses [46]. In addition, it is present in feedstuff like alfalfa, and it has become a popular human and animal supplement in the past decade [47]. The presence of this metabolite in all our study groups contrasts with the absence of this metabolite in the AH of other species. The fact that horses are hind-gut fermenters with a rich microbiome may reflect a gut–eye axis interaction, raising the possibility that gut microbial metabolism contributes to the presence of this metabolite in ocular AH. While dimethyl sulfone has not been reported in ocular biofluids in human, rats, rabbits and other species, a recent study in sheep described increasing concentrations of this metabolite in AH as time passed after death [48]. While elevated endogenous dimethyl sulfone has been associated with metabolic dysregulation and increased risk of diabetic retinopathy in humans [49], several studies have shown that exogenous supplementation of this metabolite exerts neuroprotective effects on the retina. For example, in light-induced retinal degeneration, dimethyl sulfone supplementation was found to attenuate oxidative damage and preserve retinal structure and function, including maintenance of electroretinography responses [50]. This apparent contradiction may reflect the difference between baseline accumulation of dimethyl sulfone as a biomarker of metabolic stress, versus controlled therapeutic administration as an antioxidant intervention. Our findings show the presence of endogenous dimethyl sulfone in the AH of horses, and increased concentration in the AH of eyes with cataract and chronic retinal lesions, warranting further studies focused on this compound.

### 4.3. Metabolomic Trends in Horse AH Associated with Ocular Disease

In our study, 2-hydroxivalerate showed a trend to increase in the AH of equine eyes with cataracts (Table 4). 2-hydroxivalerate is a degradation product of isoleucine, and in ocular diseases, a higher metabolic demand or oxidative stress could alter enzymatic activity and increase isoleucine degradation products such as 2-hydroxivalerate [51]. The present findings add to a growing body of evidence that altered amino acid concentrations in AH are a consistent finding in eyes with cataracts across species.

In addition, pyruvate showed a tendency to be increased in eyes with ACD compared to controls. Pyruvate has been shown to scavenge reactive oxygen species and support mitochondrial function, features particularly relevant to maintaining the health of metabolically active ocular tissues. Although this difference did not reach statistical significance, it is worth highlighting given the known role of pyruvate as a central metabolite in cellular energy metabolism and its function as a potent antioxidant [45]. Increased pyruvate levels may reflect a compensatory response to increased oxidative stress or altered glycolytic activity in diseased eyes. However, it is important to point out that our results are exploratory and these trends need to be validated in the context of larger, stratified future studies.

We found a trend toward lower creatinine concentration in the AH of the eyes affected by retinal disease compared to control eyes. Creatinine is the end product of creatine-phosphate and creatine (an organic acid naturally produced in the body). This metabolite has anti-inflammatory, antioxidant, and immunomodulatory effects [52] and AH creatinine concentrations closely mirror those in plasma [53]. To our knowledge there are few bibliographical references linking altered creatinine levels in AH of eyes with retinal lesions. Two eyes from the retina group showed retinal dysplasia, and four eyes had bullet-hole lesions of varying degrees. Retinal dysplasia is an uncommon congenital malformation of the neurosensory retina. It can be secondary to an intrauterine infection or be associated with congenital ocular anomalies [54]. Bullet-hole chorioretinitis involves small, multifocal lesions located ventral to the optic disc, appearing white with pigment in the center [55]. The etiology and clinical significance of bullet-hole chorioretinopathy is not completely understood. This finding has been related to previous experimental equine herpes virus type 1 infection and respiratory disease in previous studies [56,57]. None of the horses included in the study presented signs compatible with viral infection or evidence of active chorioretinal inflammation. Both dysplasia and bullet-hole lesions are considered non-progressive, and it is likely that these changes occurred before the time of sample collection.

In this study we applied univariable mixed-effects model to explore metabolite differences between groups. This was required due to our sampling design, in which a proportion of the horses contributed both the right and left eye, while others contributed just one eye to the study. Mixed models provide a robust statistical framework to address this.

Although some of the observed changes did not meet statistical significance, they could be biologically meaningful and support trends in other species. Given the exploratory nature of the study, these results may assist hypothesis generation and prioritize future targets in metabolomic studies of equid AH. Due to the difficulties derived from obtaining very fresh AH samples from horses and donkey eyes, resulting in small to moderate sample size, we adopted a lenient inclusion threshold (raw *p*-values < 0.2) for FDR correction. This approach is common when the goal is to detect biologically significant signals that may be masked by the conservative nature of multiple corrections. Using a broader selection of raw *p*-values enabled us to include borderline signals that may become clearer in validation studies.

#### Limitations

Considering that this was a convenience sample, AH samples were gathered over a period of 2 years rather than during a specific season, which may have contributed to variability. Furthermore, ACD and retinal diseases were somewhat heterogeneous, and groups were relatively small. This reflects the opportunistic nature of sample collection and the rarity of obtaining suitable AH specimens in these species. While this limits the statistical power of group comparisons, it is an expected constraint in exploratory studies of this kind. These findings should, therefore, be interpreted cautiously and used to inform the design of future studies with larger, more balanced, and stratified cohorts.

The total number of animals, particularly donkeys, was limited because this was a convenience sample study.

This exploratory study was conducted under ethical and logistical constraints given the limited availability of AH samples from equine clinical cases, leading to modest group sizes and inherent heterogeneity across groups. In addition, samples were collected within five minutes postmortem, and the lack of control over seasonal timing (an unavoidable consequence of the opportunistic sampling strategy) may have introduced additional variability into the metabolomic profiles. This factor should be considered when interpreting the findings. The control group included individuals of both sexes and a range of ages, reflecting the diversity commonly encountered in clinical and opportunistic sampling scenarios. While this introduces a degree of biological variability, it also enhances the real-world applicability of the findings. For the purposes of this exploratory study, the control group was analyzed as a single reference cohort to enable comparison with diseased groups. While we performed limited statistical checks for age and sex effects, these were not the focus of the study and should be interpreted with caution due to small subgroup sizes. Importantly, the heterogeneity of the control group is acknowledged as a limitation, and readers are cautioned to interpret the findings within the appropriate context. Future studies with larger, stratified cohorts will be needed to disentangle the effects of sex and age on AH metabolomic profiles.

All spectra (including controls and diseased eyes from both horses and donkeys) were obtained using identical acquisition and processing protocols. It is worth mentioning that the NOESY experiments, although acquired for every sample, were not employed in this study due to poor water suppression signals obtained in some samples, which made appropriate phase and baseline correction difficult close to the water signal. This complication hindered accurate signal fitting with Chenomx using the NOESY experiment. Alternatively, we used the CPMG pulse sequence, which, in our hands, provided better water suppression performance. It is important to note that the Chenomx signature library is built using NOESY spectra, not the CPMG pulse sequence. As a result, the absolute concentration values may not be directly comparable to studies using the NOESY experiment (concentrations obtained using the CPMG experiment are expected to be lower due to T2 signal filtering). However, the relative differences among groups are considered valid and can be compared to other studies because the same acquisition and processing protocols were consistently applied to all samples in this study.

## 5. Conclusions

This study provides a foundational metabolomic characterization of AH in horses and donkeys using 1H-NMR spectroscopy. The overall metabolic fingerprint was largely conserved between the two species. Additionally, variations in metabolite concentrations were identified in the eyes of horses with and without different ocular disorders, suggesting that these metabolites may have potential as preliminary biomarkers for ocular diseases in horses. Differences related to age and sex were also noted. These findings highlight the feasibility and value of metabolomic fingerprinting in veterinary ophthalmology and underscore the need for further studies with larger, stratified cohorts to assess the diagnostic and prognostic value of specific metabolites in ocular disease in equids and other veterinary species.

## Figures and Tables

**Figure 1 animals-15-02810-f001:**
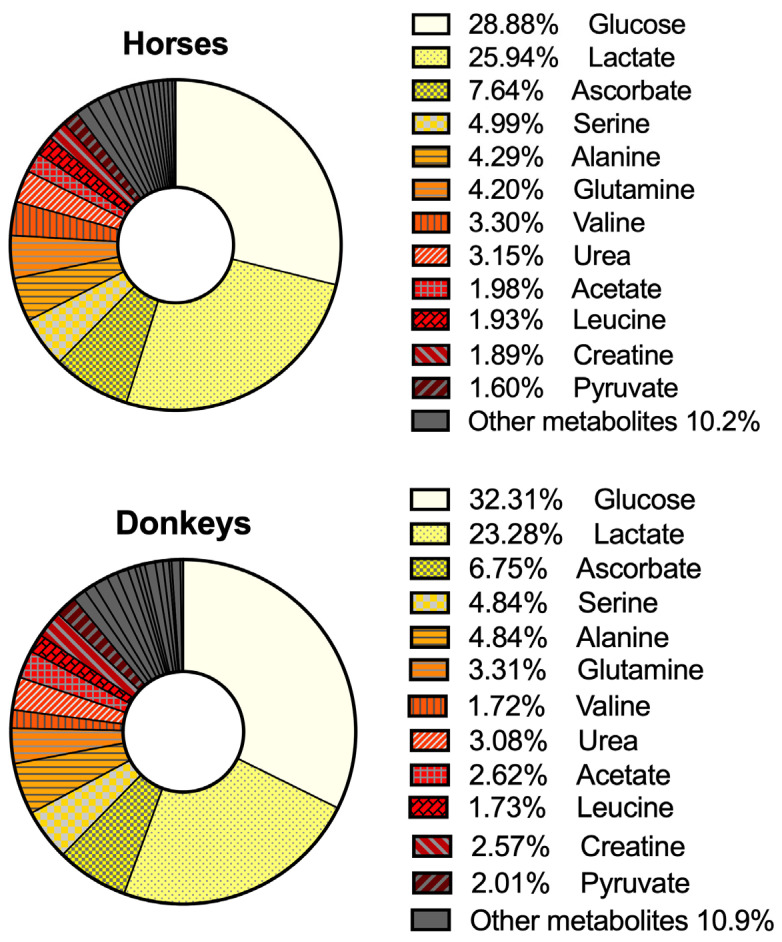
Proportional distribution of quantified metabolites in the aqueous humor (AH) of horses and donkeys. Pie chart illustrates the relative contribution of each of the 27 quantified metabolites to the total sum of metabolite concentrations identified in each species. Each metabolite is expressed as a proportion of the total metabolite pool in each species. This visualization highlights differences in overall metabolic profile balance, even when absolute concentrations may appear similar between groups. Other metabolites with lower relative concentrations in AH include arginine, citrate, creatine, dimethyl sulfone, ethanol, isoleucine, methanol, phenylalanine, proline, propylene glycol, threonine, tryptophan, tyrosine, histidine and 2-hydroxyvalerate. Note: only the most abundant metabolites are included; AH contains additional undetected compounds.

**Figure 2 animals-15-02810-f002:**
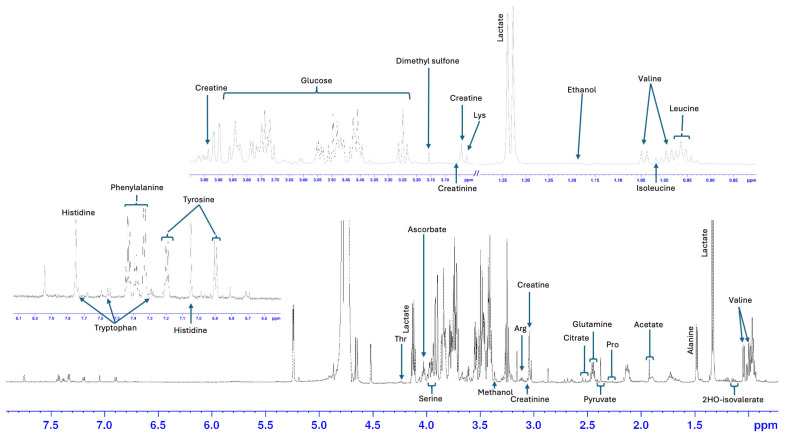
Representative nuclear magnetic resonance (NMR) 600 MHz spectrum and expanded areas of the AH of a horse’s healthy eye. Assigned metabolites are based on Chenomx database Software. Lys: Lysine, Thr: Threonine, Arg: Arginine, Pro: Proline.

**Figure 3 animals-15-02810-f003:**
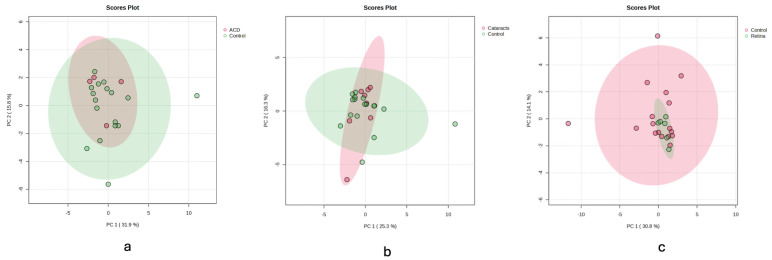
Principal component analysis (PCA) score plot comparing metabolomic profiles of equine AH samples of the (**a**) cataract, (**b**) retina and (**c**) anterior chamber disease (ACD) groups versus controls. The PCA was used solely to explore the overall data structure. Each point represents a sample projected onto the first two principal components (PC1 and PC2), which together explain varying percentages of the variance. Ellipses represent 95% confidence regions. This exploratory multivariate analysis did not serve as a confirmatory tool and was not used to derive statistical significance.

**Table 1 animals-15-02810-t001:** Horse ocular lesions found in each group.

Group	Lesion	N
Cataract group	Incipient cataract	5
	Immature cataract	2
	Mature cataract	1
Retina group	<10 bullet holes	2
	>10 bullet holes	2
	Retinal dysplasia	2
ACD group	Anterior uveitis	3
	Corpora nigra cyst	1

ACD: anterior chamber disease.

**Table 2 animals-15-02810-t002:** Median and interquartile range (IQR) values for each metabolite (mM) identified in horses and donkeys AH without ocular pathology.

	Horses			Donkeys		
Metabolite	Median	IQR	N	Median	IQR	N
Acetate	0.30	0.09	16	0.29	0.08	5
Alanine	0.65	0.12	17	0.80	0.19	5
Arginine	0.12	0.04	17	0.17	0.06	5
Ascorbate	1.16	0.70	17	1.11	0.18	5
Citrate	0.11	0.05	17	0.11	0.02	5
Creatine	0.29	0.13	17	0.33	0.13	5
Creatinine	0.07	0.04	17	0.09	0.08	5
Dimethyl sulfone	0.07	0.11	12	0.04	0.01	5
Ethanol	0.17	0.29	14	0.23	0.06	4
Glucose	4.37	1.68	17	5.32	0.76	5
Glutamine	0.64	0.34	17	0.55	0.31	5
Isoleucine	0.18	0.06	17	0.17	0.05	5
Lactate	3.92	2.15	17	3.84	0.39	5
Leucine	0.29	0.10	17	0.43	0.09	5
Methanol	0.11	0.21	17	0.08	0.09	5
Phenylalanine	0.19	0.07	17	0.20	0.03	5
Proline	0.07	0.04	14	0.14	0.06	2
Propylene glycol	0.08	0.05	17	0.12	0.03	5
Pyruvate	0.24	0.06	16	0.34	0.09	5
Serine	0.76	0.30	12	0.72	0.56	4
Threonine	0.09	0.09	12	0.16	0.05	5
Tryptophan	0.04	0.03	16	0.04	0.02	4
Tyrosine	0.15	0.04	17	0.16	0.02	5
Urea	0.48	0.21	13	0.39	0.23	5
Valine	0.50	0.37	17	0.25	0.55	5
Histidine	0.10	0.03	17	0.08	0.03	5
2-Hydroxyvalerate	0.01	0.003	8	0.004	0.002	4

**Table 3 animals-15-02810-t003:** Statistically significant differences in metabolite concentration between eye disease and control groups based on false discovery rate (FDR)-adjusted *p*-values < 0.05, expressed as median and IQR metabolite concentrations (mM).

Metabolite	Group	Median	IQR	Group	Median	IQR	FDR-Adj. *p*
Arginine	Retina	0.07	0.01	Control	0.12	0.07	0.05
Dimethyl sulfone	Retina	0.28	0.21	Control	0.07	0.05	0.00
	Cataract	0.16	0.23	Control	0.07	0.05	0.05
Valine	Retina	0.15	0.07	Control	0.50	0.33	0.03

IQR: interquartile range. FDR-Adj. *p*: False Discovery Rate-Adjusted *p* value.

**Table 4 animals-15-02810-t004:** Observed trends towards differences in metabolite concentration between eye disease and control groups based on FDR-adjusted *p*-values < 0.2, expressed as median and IQR metabolite concentrations (mM).

Metabolite	Group	Median	IQR	Group	Median	IQR	FDR-Adj. *p*
Arginine	Cataract	0.08	0.06	Control	0.12	0.07	0.09
Ascorbate	ACD	1.03	1.02	Control	1.16	0.70	0.12
	Retina	0.79	0.14	Control	1.16	0.70	0.12
Creatinine	Retina	0.04	0.02	Control	0.48	0.27	0.15
2-hydroxivalerate	Cataract	0.02	0.01	Control	0.01	0.00	0.07
Pyruvate	ACD	0.38	0.34	Control	0.24	0.06	0.09
Valine	ACD	0.31	0.20	Control	0.50	0.33	0.10

ACD: anterior chamber disease. IQR: interquartile range. FDR-Adj. *p*: False Discovery Rate-Adjusted *p* value.

## Data Availability

The original contributions presented in this study are included in the article. Further inquiries can be directed to the corresponding author.

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
