# Peer review of "Exploratory Metabolomic Fingerprinting of Aqueous Humor in Healthy Horses and Donkeys, and in Horses with Ocular Pathologies"

_animals, 2025, doi:10.3390/ani15192810_

Round 1
Reviewer 1 Report (New Reviewer)
Comments and Suggestions for Authors
This article is very meaningful. The overall quality of the research on the ocular pathology of horses and donkeys is good. However, there are still several issues that need to be discussed with the author: 1. Can the number of donkeys be increased? 2. Investigate whether the animals have any relevant medical history? What is the nutritional level like for them? 3. Among the 24 horses, how many are stallions and how many are mares? 4. The references for the past three years are relatively few. Could you please add more? 5. The text in the result graphs is too small, such as in Figure 3, etc.
Author Response
REVIEWER 1
This article is very meaningful. The overall quality of the research on the ocular pathology of horses and donkeys is good. However, there are still several issues that need to be discussed with the author: 1. Can the number of donkeys be increased? 2. Investigate whether the animals have any relevant medical history? What is the nutritional level like for them? 3. Among the 24 horses, how many are stallions and how many are mares? 4. The references for the past three years are relatively few. Could you please add more? 5. The text in the result graphs is too small, such as in Figure 3, etc.
- Can the number of donkeys be increased?
ANSWER: Unfortunately, the number of donkeys cannot be increased. In accordance with the comments of reviewer 2, we have removed the donkeys from the statistical analysis. Given that there is no information in the literature regarding the metabolomic composition of aqueous humor in this species, we have decided to keep the information in the manuscript and report this simply as descriptive data without conducting formal statistical comparison of horses vs. donkeys (Figure 1, table 2, lines 201-202).
- Investigate whether the animals have any relevant medical history? What is the nutritional level like for them?
ANSWER: The animals included in this study were acquired from a single, authorized supplier, who sourced them from various external facilities not affiliated with the University. As described in lines101-104: “Only animals that were not receiving any systemic or topical medication were enrolled. In addition, all animals were appropriately vaccinated and dewormed in accordance with standard protocols, as these were prerequisites for admission and continued housing in the facilities”. We have added that: All animals enrolled in the study presented a body condition score between 4 and 6 out of 9 (lines 91-92).
- Among the 24 horses, how many are stallions and how many are mares?
ANSWER: Regarding sex distribution, 8 were control eyes from females and 9 were control eyes from male (5 eyes from geldings and 4 from stallions) (Lines 199-201).
- The references for the past three years are relatively few. Could you please add more?:
ANSWER: The following recent references have been added to the bibliography:
Huo, Q., Xu, Y., Wang, Y., Zhang, S., Liu, Z., & Li, J. (2025). Metabolomics analysis of aqueous humor from patients with high-myopia complicated nuclear cataract. Frontiers in Medicine, 12, 1454840
Monu, M., Kumar, B., Asfiya, R., Nassiri, N., Patel, V., Das, S., ... & Singh, P. K. (2025). Metabolomic Profiling of Aqueous Humor From Glaucoma Patients Identifies Metabolites With Anti-Inflammatory and Neuroprotective Potential in Mice. Investigative Ophthalmology & Visual Science, 66(5), 28-28
Qi, X., Dai, Y., Pan, X., Shan, X., Ge, Q., Zhou, J., ... & Lan, J. (2025). Oleic acid association with primary angle-closure glaucoma: A finding using metabolomics. Experimental Eye Research, 110418
- The text in the result graphs is too small, such as in Figure 3, etc:
ANSWER: Figure 3 has been modified to enlarge the text size and improve its readability
Reviewer 2 Report (New Reviewer)
Comments and Suggestions for Authors
This paper details the inaugural investigation into the metabolic composition of aqueous humor (AH) in horses and donkeys, utilizing proton nuclear magnetic resonance (^1H-NMR) spectroscopy. Its primary aims were to establish a fundamental metabolic profile for healthy equids and to pinpoint metabolic deviations associated with various ocular pathologies.
A total of 40 AH specimens were analyzed, comprising 35 from horses and 5 from donkeys. These were procured from 24 horses and 3 donkeys, all slated for slaughter and maintained under consistent housing conditions before sample acquisition. The equine samples were categorized into several groups: 17 healthy controls (further broken down by age – 8 young, 9 aged 16 or older – and sex – 7 females, 10 males), 8 eyes with cataracts, 6 with retinal disorders (including bullet-hole chorioretinitis or retinal dysplasia), and 4 exhibiting anterior chamber diseases (such as uveitis or corpora nigra cysts). Additionally, 5 healthy donkey eyes were included. Standardized protocols were followed for NMR spectra acquisition, and metabolite quantification was performed using Chenomx software. This process led to the identification of 27 distinct metabolites, encompassing amino acids, their derivatives, organic acids, saccharides, and other low-molecular-weight compounds.
While most metabolite concentrations showed comparable levels between species, a few significant differences emerged based on species, age, and sex. Specifically, leucine levels were notably elevated in donkeys (p=0.035), threonine was found in higher concentrations in younger horses (p=0.037), and creatinine was more abundant in male equids (p=0.036). Among diseased eyes, the group suffering from retinal pathology exhibited significantly reduced levels of both arginine (0.070 ± 0.010 mM/L compared to 0.120 ± 0.065 in controls; p=0047) and valine (0.147 ± 0.069 vs. 0.499 ± 0.328; p=0.029) relative to healthy controls. Conversely, dimethyl sulfone concentrations were markedly elevated in both the retinal disease cohort (0.281 ± 0.208 vs. 0.066 ± 0.052; p=0.004) and the cataract group (0.164 ± 0.232 vs. 0.066 ± 0.052; p=0.050) when compared to controls. Other observed patterns, though not statistically significant after false discovery rate correction, included diminished ascorbate in eyes with anterior chamber disease and retinal pathology, lower creatinine in the retina group, and elevated 2-hydroxivalerate in cataractous eyes. Multivariate statistical techniques (such as PCA, PLS-DA, and random forest) did not yield clear or robust distinctions between the different groups, reinforcing the preliminary and exploratory character of this dataset.
The study's discussion interprets the decreased arginine and valine as potential indicators of oxidative stress and cataract progression. Elevated dimethyl sulfone is speculatively linked to interactions within the gut-eye axis, and variations in threonine are attributed to age-related protein metabolism. The authors candidly acknowledge several limitations, including the restricted and imbalanced group sizes, the diverse nature of the disease subgroups, the use of convenience sampling, and the technical choice of CPMG over NOESY sequencing, which might influence absolute metabolite quantification but not relative comparisons. Ultimately, the research establishes the viability of AH metabolomic analysis in equids, uncovering both conserved metabolic profiles and distinctions related to species, sex, and age. It proposes initial candidate biomarkers for eye diseases but strongly highlights the necessity for more extensive, stratified studies to validate these findings and translate them into clinical applications.
While this manuscript breaks new ground by offering the first detailed metabolic analysis of equid aqueous humor, its conclusions are significantly hampered by several methodological and design shortcomings. The study's design is undermined by notably small and disproportionate sample sizes; for instance, only five healthy donkey samples were included. Furthermore, the equine disease categories lack homogeneity, and the reliance on convenience sampling from animals destined for slaughter introduces an inherent bias. The broad demographic range (sex and age) within the control group also contributes to considerable biological variability.
From a methodological standpoint, the choice of the CPMG sequence over NOESY limits the generalizability and cross-study comparability of the metabolomic data. The statistical analysis raises concerns due to the presentation of "trends" based on raw p-values (some as high as < 0.2) as potentially meaningful findings, which increases the likelihood of false positives. Moreover, the application of complex multivariate statistical models (like PCA, PLS-DA, and random forest) provided minimal substantiative insight, primarily due to the insufficient sample sizes.
Regarding the results, while 27 metabolites were identified, numerous reported group differences appear marginal. The clarity of some figures and tables is compromised by a cluttered presentation, and there is an inconsistent distinction between statistically significant discoveries and more exploratory observations. The discussion section frequently engages in speculation, such as positing a 'gut-eye axis' connection for dimethyl sulfone or interpreting creatinine variations without firm evidence. Hypotheses are often presented with an unwarranted degree of certainty, bordering on definitive conclusions. While the study's pioneering nature in equine AH metabolomics is undisputed, its actual contribution remains predominantly descriptive, and any assertions of biomarker discovery are premature without more extensive, targeted validation efforts.
Although the authors do list limitations, these crucial caveats—particularly concerning sample size, group heterogeneity, and potential postmortem or seasonal influences on metabolite profiles—warrant more prominent and earlier emphasis. The identified metabolic signals (e.g., arginine, valine, dimethyl sulfone) are intriguing as preliminary observations, but the study’s statistical power is demonstrably weak, necessitating extremely cautious interpretation of any conclusions. The enthusiasm for deploying multiple advanced statistical models was not adequately supported by the available sample size. The discussion, while offering good contextualization, leans too heavily into speculative interpretations and frames unsupported findings as more conclusive than warranted, underscoring the need to more robustly characterize the study as purely exploratory. Delete donkey from analysis.
Author Response
Reviewer 2
This paper details the inaugural investigation into the metabolic composition of aqueous humor (AH) in horses and donkeys, utilizing proton nuclear magnetic resonance (^1H-NMR) spectroscopy. Its primary aims were to establish a fundamental metabolic profile for healthy equids and to pinpoint metabolic deviations associated with various ocular pathologies.
A total of 40 AH specimens were analyzed, comprising 35 from horses and 5 from donkeys. These were procured from 24 horses and 3 donkeys, all slated for slaughter and maintained under consistent housing conditions before sample acquisition. The equine samples were categorized into several groups: 17 healthy controls (further broken down by age – 8 young, 9 aged 16 or older – and sex – 7 females, 10 males), 8 eyes with cataracts, 6 with retinal disorders (including bullet-hole chorioretinitis or retinal dysplasia), and 4 exhibiting anterior chamber diseases (such as uveitis or corpora nigra cysts). Additionally, 5 healthy donkey eyes were included. Standardized protocols were followed for NMR spectra acquisition, and metabolite quantification was performed using Chenomx software. This process led to the identification of 27 distinct metabolites, encompassing amino acids, their derivatives, organic acids, saccharides, and other low-molecular-weight compounds.
While most metabolite concentrations showed comparable levels between species, a few significant differences emerged based on species, age, and sex. Specifically, leucine levels were notably elevated in donkeys (p=0.035), threonine was found in higher concentrations in younger horses (p=0.037), and creatinine was more abundant in male equids (p=0.036). Among diseased eyes, the group suffering from retinal pathology exhibited significantly reduced levels of both arginine (0.070 ± 0.010 mM/L compared to 0.120 ± 0.065 in controls; p=0047) and valine (0.147 ± 0.069 vs. 0.499 ± 0.328; p=0.029) relative to healthy controls. Conversely, dimethyl sulfone concentrations were markedly elevated in both the retinal disease cohort (0.281 ± 0.208 vs. 0.066 ± 0.052; p=0.004) and the cataract group (0.164 ± 0.232 vs. 0.066 ± 0.052; p=0.050) when compared to controls. Other observed patterns, though not statistically significant after false discovery rate correction, included diminished ascorbate in eyes with anterior chamber disease and retinal pathology, lower creatinine in the retina group, and elevated 2-hydroxivalerate in cataractous eyes. Multivariate statistical techniques (such as PCA, PLS-DA, and random forest) did not yield clear or robust distinctions between the different groups, reinforcing the preliminary and exploratory character of this dataset.
The study's discussion interprets the decreased arginine and valine as potential indicators of oxidative stress and cataract progression. Elevated dimethyl sulfone is speculatively linked to interactions within the gut-eye axis, and variations in threonine are attributed to age-related protein metabolism. The authors candidly acknowledge several limitations, including the restricted and imbalanced group sizes, the diverse nature of the disease subgroups, the use of convenience sampling, and the technical choice of CPMG over NOESY sequencing, which might influence absolute metabolite quantification but not relative comparisons. Ultimately, the research establishes the viability of AH metabolomic analysis in equids, uncovering both conserved metabolic profiles and distinctions related to species, sex, and age. It proposes initial candidate biomarkers for eye diseases but strongly highlights the necessity for more extensive, stratified studies to validate these findings and translate them into clinical applications.
While this manuscript breaks new ground by offering the first detailed metabolic analysis of equid aqueous humor, its conclusions are significantly hampered by several methodological and design shortcomings. The study's design is undermined by notably small and disproportionate sample sizes; for instance, only five healthy donkey samples were included. Furthermore, the equine disease categories lack homogeneity, and the reliance on convenience sampling from animals destined for slaughter introduces an inherent bias. The broad demographic range (sex and age) within the control group also contributes to considerable biological variability.
ANSWER: In accordance with Reviewer’s 2 recommendation, the aqueous humor samples from donkeys were excluded from the statistical analysis due to a limited sample size. To the authors' knowledge, no studies have been published to date regarding the metabolomic composition of aqueous humor in this species. Therefore, we considered it scientifically relevant to retain this information in the manuscript. We agree that it is possible that the sampling method may have introduced a bias. However, obtaining aqueous humor samples from clinically healthy horses is would not be ethical, and obtaining samples from eyes with different diseases particularly challenging, as this species present specific challenges for invasive sampling procedures.
From a methodological standpoint, the choice of the CPMG sequence over NOESY limits the generalizability and cross-study comparability of the metabolomic data.
ANSWER: We appreciate the reviewer’s comment and the opportunity to clarify our rationale for selecting the CPMG pulse sequence over NOESY. While it is true that NOESY is often used in metabolomics and is the default acquisition method for some spectral libraries (e.g., Chenomx or BAYESIL), the protocolary use of such advanced data bases requires sample filtration.
It is important to emphasize that our samples consisted of unfiltered aqueous humor, collected in very limited volumes, therefore physical filtering and sample manipulation could be a source of sample loss. Although we did acquire NOESY spectra for every sample, we observed inconsistent water suppression and phase instability in several cases, particularly in proximity to the water peak. Furthermore, we observed interfering broad peaks likely due to the presence of proteins and lipids in the unfiltered matrix. These issues hindered reliable metabolite fitting using the Chenomx software, which relies on clear baselines and accurate phasing for quantification.
In contrast, the CPMG sequence, with its Tâ‚‚ filtering capabilities, is specifically designed to suppress broad signals from proteins, lipids, and other macromolecules (a clear advantage when analyzing unfiltered biofluids such as was the case with our aqueous humor samples). In our hands, CPMG yielded higher-quality, more interpretable spectra across all samples, enabling robust relative comparisons.
While we acknowledge that using CPMG may limit direct absolute comparisons with other studies that use NOESY acquired concentrations in particular, this limitation is clearly stated in the manuscript, however our result will be directly comparable to studies using CPMG sequence which is standard too. Crucially, because we applied the same acquisition and processing parameters across all groups, the within-study comparisons remain valid and unaffected by this choice.
Finally, we note that CPMG is a widely accepted and standard pulse sequence in untargeted metabolomics (particularly when working with small-volume, protein-containing, unfiltered fluids) which was exactly our situation. Its use is well-documented in literature, including in major book references such as:
* Lindon et al. (2007). The Handbook of Metabonomics and Metabolomics, Elsevier.
* Emwas et al. (2019). NMR Spectroscopy for Metabolomics Research, Metabolites, 9(7), 123.
Therefore, given the nature of our biological samples, the necessity to use unfiltered samples due to small volume of the samples, and study aims, we believe CPMG was the most appropriate and technically sound choice.
The statistical analysis raises concerns due to the presentation of "trends" based on raw p-values (some as high as < 0.2) as potentially meaningful findings, which increases the likelihood of false positives. Moreover, the application of complex multivariate statistical models (like PCA, PLS-DA, and random forest) provided minimal substantiative insight, primarily due to the insufficient sample sizes.
ANSWER: The presentation of our results of statistical analysis was improved for clarity in the result and discussion sections, and several references to the exploratory nature of this study have been added throughout the manuscript (Lines 20-26, 39-42, 158-159, 336, 358) to ensure all readers are aware of the exploratory characteristics and to prevent them from drawing firm conclusions about the results reported. The changes introduced to the manuscript clearly report (1) metabolites with a FDR adjusted p-value <0.05 (see lines 175-178 and Table 3) and paragraph under “Significant metabolite changes in horse eyes with cataracts and retinal disease” subheading within the Discussion (Line 340); and (2) metabolites with a FDR adjusted p-value <0.2 (see Table 4) and paragraph under “Metabolomic trends in horse AH associated with ocular disease” subheading within the Discussion (line 384).
The likelihood of false positive findings is minimized in this study by using the “multiple testing corrections of the Benjamini-Hochberg false discovery rate method” as described in the materials and methods section (see lines 175-178).
Therefore, we believe we now clearly state our methods, clearly separate those findings with a statistical significance below 0.05 FDR-adjusted p-values from those that could be considered “trends” with a FDR-adjusted p-value <0.2 (or q value <0.2), and have clearly explained that we have adjusted for the possibility of false positives associated with multiple comparisons by using the Benjamini-Hochberg FDR method.
We also would like to mention that in other exploratory metabolomic analysis a similar statistical approach has been used, such as in:
Islam, S. J., Liu, C., Mohandas, A. N., Rooney, K., Nayak, A., Mehta, A., & Searles, C. D. (2024). Metabolomic signatures of ideal cardiovascular health in black adults. Scientific Reports, 14(1), 1794
Aredo, J. V., Purington, N., Su, L., Luo, S. J., Diao, N., Christiani, D. C., & Han, S. S. (2021). Metabolomic profiling for second primary lung cancer: A pilot case-control study. Lung cancer, 155, 61-67
Finally, we improved clarity regarding the more complex multivariate statistical modelling, and clearly explain that this was performed in order to ensure that findings of univariate analysis were not influenced by some data clustering (Lines 256-261 and Fig 3 legend). Our limitation of a relatively small number of samples (compared to laboratory animal studies) is acknowledged and included in our discussion and limitations.
Regarding the results, while 27 metabolites were identified, numerous reported group differences appear marginal. The clarity of some figures and tables is compromised by a cluttered presentation, and there is an inconsistent distinction between statistically significant discoveries and more exploratory observations.
ANSWER: Following Reviewer 2 and 3, Figure 3 has been modified to improve its readability.
The discussion section frequently engages in speculation, such as positing a 'gut-eye axis' connection for dimethyl sulfone or interpreting creatinine variations without firm evidence. Hypotheses are often presented with an unwarranted degree of certainty, bordering on definitive conclusions. While the study's pioneering nature in equine AH metabolomics is undisputed, its actual contribution remains predominantly descriptive, and any assertions of biomarker discovery are premature without more extensive, targeted validation efforts.
Although the authors do list limitations, these crucial caveats—particularly concerning sample size, group heterogeneity, and potential postmortem or seasonal influences on metabolite profiles—warrant more prominent and earlier emphasis. The identified metabolic signals (e.g., arginine, valine, dimethyl sulfone) are intriguing as preliminary observations, but the study’s statistical power is demonstrably weak, necessitating extremely cautious interpretation of any conclusions. The enthusiasm for deploying multiple advanced statistical models was not adequately supported by the available sample size. The discussion, while offering good contextualization, leans too heavily into speculative interpretations and frames unsupported findings as more conclusive than warranted, underscoring the need to more robustly characterize the study as purely exploratory. Delete donkey from analysis.
ANSWER: We thank the reviewer for this important observation. In response, we have revised the discussion to better reflect the exploratory nature of the study. Speculative statements have been reframed to draw more directly from the data and to avoid overinterpretation. Where biological hypotheses are proposed, these have been clearly identified as preliminary and contextualized within the limitations of our sample size and design.
Furthermore, we have now emphasized more prominently that this was a post-mortem exploratory study using opportunistically collected samples, and that seasonal variation and disease heterogeneity could not be fully controlled. The need for larger, confirmatory, and stratified studies has been reiterated more explicitly throughout. Additionally, we have clarified that the intention behind including advanced multivariate methods such as PCA was strictly exploratory and intended to assess structural features of the dataset rather than serve as confirmatory evidence.
Following Reviewer 1 and 2 suggestions, we have removed the formal statistical comparison between donkey and horse metabolomic data. We considered we should preserve the donkey data only as descriptive because there are no other similar studies in this species.
Reviewer 3 Report (New Reviewer)
Comments and Suggestions for Authors
The paper is devoted to metabolomic analysis of aqueous humor (AH) of horses and donkeys. The main goals of the work were to establish baseline levels of major metabolites present in AH and to compare AH metabolomic profiles of animals belonging to related species (horses and donkeys), and also to evaluate metabolomic differences between healthy horses and horses with eye diseases. To achieve these goals, the authors collected an impressive collection of 45 AH samples and performed their quantitative metabolomic profiling using NMR. Sample collection, NMR measurements and NMR data treatment were performed at a good scientific level. However, the statistical analysis and presentation of the results raise some questions.
- There are inconsistencies between Figure 1 and Table 2. For example, the level of urea in Figure 1 is 3.15% for horses and 0.32% for donkeys (tenfold difference); in Table 1 the concentrations of urea are 0.477 mmol/L and 0.389 mmol/L (20% difference). For leucine, the data in Figure 1 are 1.93% and 1.78%, while in Table 2 the concentrations are 0.292 mmol/L and 0.434 mmol/L. The authors should carefully check all data presented in Figure 1 and Table 2.
- Figure 2: Table 1 contains data on 27 metabolites, while in Figure 2 only 19 metabolites are assigned. Please assign the remaining compounds.
- Figure 3: Only panels b) and c) carry scientifically important information, other panels should be either removed or shifted to SI.
- Figure 3b and 3c: PCA was performed for all samples divided into four groups. First, it is unclear which color corresponds to which group. Second, combining all samples in one plot is not a good idea, since dispersion between, say, horses and donkeys may hide the differences between healthy and retina equine groups. Third, using the PLS-DA model can be useful if the difference between groups is already visible in the PCA score plot and this difference needs to be emphasized. This is obviously not the case. My suggestions: make separate PCA plots for following pairs: 1) horses vs. donkeys; 2) cataract vs. control; 3) retina vs. control; 4) ACD vs. control. These four plots will provide significantly more useful information than what is currently shown in Figure 3.
- Analysis of age-dependent differences: the authors divided all equine samples in two groups (below 16 y.o. and above 16 y.o.) and compared these two groups. IMO, approach that is more productive would be to plot the concentration of each compound against age, and look for correlations between age and metabolite concentrations.
- It would be very useful if the author make an Excel table with metabolite concentrations for every sample and either place this table in SI or depose in a public database.
There also several minor mistakes:
Lines 210-211: histidine and creatine are amino acids, not amino acid derivatives. 2-Hydroxyisovalerate is an organic acid.
Table 3 – to name a group, use either “retina” or “retinal”, but not both.
Line 395 – replace “Two-hydroxivalerate” with “2-hydroxyvalerate”
Table 2: the concentrations are expressed in mM/L. There is no such unit. Use either mM or mmol/L.
Lines 254-259 – add units to the concentration values.
Altogether, the author obtained interesting experimental data, but the analysis of these data needs significant improvement. Paper can be published in Animals only after major revision.
Author Response
Reviewer 3
- There are inconsistencies between Figure 1 and Table 2. For example, the level of urea in Figure 1 is 3.15% for horses and 0.32% for donkeys (tenfold difference); in Table 1 the concentrations of urea are 0.477 mmol/L and 0.389 mmol/L (20% difference). For leucine, the data in Figure 1 are 1.93% and 1.78%, while in Table 2 the concentrations are 0.292 mmol/L and 0.434 mmol/L. The authors should carefully check all data presented in Figure 1 and Table 2.
ANSWER: We thank the reviewer for carefully examining the metabolite proportion data and for pointing out the large discrepancy in the proportional representation of urea between species in the pie chart. Upon re-evaluation, we identified a data entry error in the urea concentration used for the figure due to a decimal point misplacement, the urea concentration for donkeys was inadvertently entered with one order of magnitude lower than the actual value. We sincerely apologize for this oversight. The figure has been redone using corrected concentration data, and the updated pie chart now reflects the correct proportional contribution of urea to the sum of metabolites identified in both species.
Regarding leucine, the reviewer is right to note that while donkeys exhibited higher absolute concentrations of this metabolite (as shown in the concentration tables), leucine represented a lower proportion of the total quantified metabolome in donkeys compared to horses. This highlights an important distinction between absolute concentration and relative distribution, which we now explain more clearly in the revised figure legend and main text.
We have carefully rechecked the data for all other metabolites included in the pie chart, and no additional errors were identified. Only urea was affected by the input mistake, and this has now been fully corrected.
- Figure 2: Table 1 contains data on 27 metabolites, while in Figure 2 only 19 metabolites are assigned. Please assign the remaining compounds.
ANSWER: The missing compounds have been added to the figure. Urea appears very low in the 6ppm region in this sample and has not been annotated.
Figure 3: Only panels b) and c) carry scientifically important information, other panels should be either removed or shifted to SI.
ANSWER: Following Reviewer 3 suggestions Figure 3 has been simplified to contain relevant PCA plots only.
- Figure 3b and 3c: PCA was performed for all samples divided into four groups. First, it is unclear which color corresponds to which group. Second, combining all samples in one plot is not a good idea, since dispersion between, say, horses and donkeys may hide the differences between healthy and retina equine groups. Third, using the PLS-DA model can be useful if the difference between groups is already visible in the PCA score plot and this difference needs to be emphasized. This is obviously not the case. My suggestions: make separate PCA plots for following pairs: 1) horses vs. donkeys; 2) cataract vs. control; 3) retina vs. control; 4) ACD vs. control. These four plots will provide significantly more useful information than what is currently shown in Figure 3.
ANSWER: PCA plots depicting individual groups compared to controls have been added to the figure.
- Analysis of age-dependent differences: the authors divided all equine samples in two groups (below 16 y.o. and above 16 y.o.) and compared these two groups. IMO, approach that is more productive would be to plot the concentration of each compound against age, and look for correlations between age and metabolite concentrations.
ANSWER: We thank the reviewer for the insightful suggestion. Indeed, plotting individual metabolite concentrations against age can provide useful exploratory insights, and we recognize the value of this approach.
In our study, we did explore the effect of age both as a continuous and categorical variable. Initially, we performed analyses using age as a continuous covariate within both general linear models (GLMs) and linear mixed-effects models (LMMs). These models included sex as a fixed effect, and where applicable, individual animal as a random effect. However, these approaches did not identify any statistically significant associations between age and metabolite concentrations.
Given the small sample size and limited statistical power, we then applied a categorical age grouping (young vs. old) based on biologically relevant thresholds. This approach did reveal a statistically significant difference for one metabolite (threonine). For this reason, and to avoid overstating patterns not supported by the data, we opted to retain age group as a categorical variable in our main analyses.
We hope this clarifies the rationale behind our approach. We agree that future studies with larger sample sizes could benefit from exploring both continuous and categorical age modelling, complemented by visualisation tools such as scatter plots with trend lines as suggested by the reviewer.
- It would be very useful if the author make an Excel table with metabolite concentrations for every sample and either place this table in SI or depose in a public database.
ANSWER: Following the Reviewer’s suggestion, an open access database will be place in an institutional repository once this manuscript is accepted for publication.
There also several minor mistakes:
Lines 210-211: histidine and creatine are amino acids, not amino acid derivatives. 2-Hydroxyisovalerate is an organic acid.
ANSWER: Creatine is often called an amino acid, however technically it is not one of the 20 amino acids that are the building blocks of all proteins.
Histidine has been corrected to amino acid and 2-Hydroxyisovalerate to organic acid within the revised manuscript.
Table 3 – to name a group, use either “retina” or “retinal”, but not both.
ANSWER: As suggested by the Reviewer, “retina” has been used in Table 3.
Line 395 – replace “Two-hydroxivalerate” with “2-hydroxyvalerate”
ANSWER: Changed as suggested by the reviewer (line 418 of revised manuscript).
Table 2: the concentrations are expressed in mM/L. There is no such unit. Use either mM or mmol/L.
ANSWER: Changed to mM as suggested by the reviewer in Tables 2-4.
Lines 254-259 – add units to the concentration values.
ANSWER: Following Reviewer 3 suggestions we hace included mM units in this section and throughout the results section.
Altogether, the author obtained interesting experimental data, but the analysis of these data needs significant improvement. Paper can be published in Animals only after major revision.
Round 2
Reviewer 2 Report (New Reviewer)
Comments and Suggestions for Authors
Thank you for response. I wish to authors much more donkey samples and looking forward to future manuscript.
Author Response
Comments 1: Thank you for response. I wish to authors much more donkey samples and looking forward to future manuscript.
Response 1: Thank you very much. If we have the opportunity to analyze additional donkey samples, we are confident that the results will be of interest and we would be pleased to share them.
Reviewer 3 Report (New Reviewer)
Comments and Suggestions for Authors
The submitted manuscript is an improved version of the previous article, and the authors' responses reflect all my comments. I have only two minor comments remaining:
- At PCA plots (Figure 3), the data for cataract and retina are not separate from the control samples, while the separation between ACD and control samples is obvious. This is quite an interesting observation that deserves further discussion. Author can evaluate which metabolites are responsible for this separation (for example, with the use of Volcano plot) and discuss the reason for increase or decrease of certain metabolites in the case of ACD.
- Authors should correct rounding throughout the text according to the rules. For example, at lines 273-279: “0.12 ± 0.06” instead of “0.120 ± 0.065”, “0.28 ± 0.21” instead of “0.281 ± 0.208”, “0.5 ± 0.3” instead of “0.499 ± 0.328”, and so on.
In my opinion, the paper can be published in Animals after minor revision.
Author Response
Comment 1: At PCA plots (Figure 3), the data for cataract and retina are not separate from the control samples, while the separation between ACD and control samples is obvious. This is quite an interesting observation that deserves further discussion. Author can evaluate which metabolites are responsible for this separation (for example, with the use of Volcano plot) and discuss the reason for increase or decrease of certain metabolites in the case of ACD.
Response 1: We thank the reviewer for this observation. Following the reviewer’s suggestion, we carefully re-examined the dataset and identified that the apparent separation of the anterior chamber disease (ACD) group in the PCA was the result of a data preparation error: three additional metabolite columns had inadvertently been included in the ACD group but not in the control group during the creation of the separate CSV file. This led to an artefactual clustering pattern.
We have corrected this error, re-performed the PCA, and confirmed that, after correction, the ACD samples do not show marked separation from controls. We have also rechecked the integrity of the cataract and retina datasets, repeated the PCA analyses, and re-generated all figures to ensure consistency across groups. As a result, the previous separation observed for the ACD group cannot be attributed to biological differences between metabolite concentrations but rather to this error. The revised manuscript now includes the corrected figure with the new PCA plot comparing ACD and controls.
Comment 2: Authors should correct rounding throughout the text according to the rules. For example, at lines 273-279: “0.12 ± 0.06” instead of “0.120 ± 0.065”, “0.28 ± 0.21” instead of “0.281 ± 0.208”, “0.5 ± 0.3” instead of “0.499 ± 0.328”, and so on.
Response 2: All data, both in the text and in the tables, have been adjusted by rounding to two decimal places.
This manuscript is a resubmission of an earlier submission. The following is a list of the peer review reports and author responses from that submission.
Round 1
Reviewer 1 Report
Comments and Suggestions for Authors
Corradini et al would like to compare by 1H-NMR the metabolome of aqueous humor between healthy horses and donkeys and horses with ocular pathologies. To do so, they consider 5 samples from donkeys and 35 from horses. Among the 35 horses, 17 were healthy, 8 with cataracts, 6 with retinal disease, and 4 with anterior chamber disease.
Each of the diseased horses were further divided according to the level of disease, so that one group was made by 5 horses, one by three and the remaining by 1-2 horses.
The healthy horses too appeared as not homogenic, because they included both male and female and spanned a wide range of ages, so that they could be divided in two further groups.
The above numbers make clear that the experimental plan was wrong.
In fact, the very very tiny number of individuals per group make most of the differences noticed as speculative, without any possibility of drawing general conclusions from the particular cases analyzed. Moreover, even if most of the groups are so tiny that the prerequisites for univariate analyses are not respected, the authors adventure themselves in multivariate analyses, even supervised, that notoriously require tenth of samples per variable.
Comments on the Quality of English LanguageThe quality of English seems adequate.
Reviewer 2 Report
Comments and Suggestions for Authors
I commend the authors for their investigation into horse ocular diseases. Although the study was very interesting, several important concerns were identified with their methods, analysis, and interpretation.
Lines 77-80. Why would you suspect differences in metabolites in the AH between sexes and between donkeys and horses. Is there any supporting literature on this hypothesis?
Lines 81-88. What was the overall health status of these animals? How did you know they were free of systemic disease? Were they vaccinated, dewormed, any bloodwork performed? Any previous or concurrent medication? All of these could potentially affect AH content.
Lines 96-99. Why is labeled as masked for review? The authors and institutions are listed on the first page
Lines 148-167. If both eyes were used in the analysis - how was the between eye correlation managed statistically. Use of data both eyes from one animal are not independent datasets and are correlated. If this is not accounted for in the analysis, then the overall number is overestimated.
Lines 163-172. Did you perform power analysis on your sample size? Based on your description, you only had three donkeys since the use of both eyes in the analysis does not represent independent samples.
Table 1. Were these conditions representing active or quiescent inflammation or uveitis?